# Characteristics and Risk of Forest Soil Heavy Metal Pollution in Western Guangdong Province, China

**Jian Kang** [1,2,3], **Xiaogang Ding** [2,*], **Hongyan Ma** [4], **Zhiming Dai** [5], **Xiaochuan Li** [2] **and Jianguo Huang** [1]

1. Key Laboratory of Vegetation Restoration and Management of Degraded Ecosystems, South China Botanical Garden, Chinese Academy of Sciences, 723 Xingke Road, Guangzhou 510650, China; kangj@scbg.ac.cn (J.K.); huangjg@scbg.ac.cn (J.H.)
2. Guangdong Academy of Forestry, 233 Guangshan First Road, Guangzhou 510520, China; lixiaochuan@sinogaf.cn
3. University of Chinese Academy of Sciences, 19(A) Yuquan Road, Beijing 100049, China
4. Guangzhou Tanhui Forestry Co., Ltd., 233 Guangshan First Road, Guangzhou 510520, China; hongyan207@163.com
5. Agricultural Science and Technology Innovation Center, 1 Nonglin Heng Road, Jiangmen 529000, China; 13822397699@139.com
* Correspondence: 27267152@sinogaf.cn

**Abstract:** West Guangdong is an important ecological barrier in Guangdong province, so understanding the spatial patterns and sources of heavy metal pollution of forest soil in this region is of great significance for ecological protection. In this study, the concentrations of heavy metals (Cd, Pb, Cu, Zn, and Ni) in forest soil were determined. Geostatistics, single-factor pollution index (PI), potential ecological risk index (RI), principal component analysis (PCA), and Pearson's correlation analysis were used to evaluate and analyze the characteristics of heavy metal pollution of forest soil. The results showed that the average concentration did not exceed the critical value. Cd, Pb, and Cu were enriched in southwest Xinxing County, while Zn and Ni were enriched in most areas of the Yunan and Yuncheng districts. Two groups of heavy metals from different sources were identified by PCA and a correlation analysis. Cd, Pb, and Cu in their respective enrichment areas were mainly from marble and cement production, whereas Zn and Ni were primarily from transportation and chemical fertilizer. Most of the study area was safe or slightly polluted while the heavy metal-enriched areas were moderately to severely polluted. The potential ecological risk was at a lower level in the study area but moderate in southwest Xinxing County. In summary, human factors impact the spatial patterns and ecological risks of heavy metals in forest soil. This study provides a scientific basis for forest soil pollution control and ecological protection.

**Keywords:** forest soil heavy metal; spatial pattern; source analysis; pollution assessment

## 1. Introduction

Forest soil is the cornerstone of the forest ecosystem and an important link between material circulation and energy flow [1]. The quality of forest soil affects the security of the ecological environment, food, water resources, etc.; thus, it has a profound impact on ecosystem functions [2,3]. However, with the rapid development of the economy, more heavy metal pollutants are entering the forest soil, leading to changes in the soil's physical and chemical properties, affecting the stability of forest ecosystems [4–6].

Heavy metals, among the many pollutants, are considered some of the most serious pollutants, e.g., they are difficult to degrade and are toxic [7,8]. Studies have shown that heavy metals in forest soil can affect the physiological activity of roots and reduce forest productivity [9,10]. These heavy metals accumulate in the trees and are transported into the food chains [11,12]. A study on post-mining areas in Eastern Slovakia found that Zn and Cd were enriched in large animals [13]. Long-term accumulation of heavy metals will

eventually endanger human health [14]. Previous studies have found that heavy metals can also get into plants and produce toxic grains [15], such as rice containing Cd [16–18]. Therefore, it is of great significance to reduce the damage of soil heavy metals by evaluating the risks of heavy metal pollutants [19,20] and analyzing the main sources of heavy metal pollutants [21,22] in forest soil.

These heavy metal pollutants mainly come from mining and light industry waste, some from cars, and even pesticides [8,23–26]. In the northern taiga forests, industrial air pollution proved to be the principal reason for the increase of Ni, Cu, and Co in the upper forest soil [27]. Dust and waste ore produced by mining are also important causes of heavy metal pollution in forest soil [28]. The studies found that Cu, Pb, Zn, and Cd content in the soil near mining areas is obviously higher than in other areas [29,30]. Moreover, a previous study showed that heavy road traffic was a vital factor that led to mosses and dwarf shrubs accumulating zinc and copper in Europe [31]. Researchers found that road traffic was the main source of Zn and Ni in fir forest soil in Poland [21]. There is another important source that cannot be ignored—the use of chemical reagents in forest management, resulting in heavy metal pollution in forest soil [32]. However, the extent and risks of heavy metal contamination need to be assessed by appropriate methods.

Previous studies on the risk assessments of heavy metal pollution have long been documented. The research included pollution assessments for specific sources [1,33]. Previous studies have shown that coal power generation has led to heavy Cd and As pollution in peripheral thermal power plants; the pollution was considered strongly contaminated in Xilinhot [26]. Holtra et al. [34] found heavy contamination around the smelter and at least moderate contamination of the soil near the smelter, based on the geoaccumulation index. Some studies have researched the ecological risks of heavy metals. For example, one study found that heavy metals had low-risk values according to the Hakanson ecological risk index in central Kenya [35]. Some studies focused on the spatial patterns of heavy metal elements. In central Europe, researchers modeled and assessed the spatial and vertical distributions of potentially toxic elements in soil based on ArcGIS-based ordinary kriging interpolation [36]. Some studies were based on the random forest model [37,38]. The Pearl River Delta, as one of the largest urban agglomerations in China, has a rich industrial system, and industrial pollution is an important factor in heavy metal pollution in this region [39,40]. Western Guangdong is also an important part of the Pearl River Delta, where mining and artificial stones are major industries. Although previous studies on heavy metals in western Guangdong detected no pollution or lower levels of pollution in farmland soil, other heavy metals with higher concentrations were found in coastal areas of western Guangdong [41]. Nevertheless, there are few works on the risks and conditions of heavy metals in forest soil in western Guangdong.

In this study, several soil samples were collected from forests in west Guangdong. In this study, we: (1) revealed the spatial patterns of heavy metals in forest soils, (2) analyzed the sources of heavy metal pollution in forest soils, and (3) evaluated the spatial patterns of pollution levels and potential ecological risks.

## 2. Materials and Methods

### 2.1. Study Area

Yunfu city is located in the west of Guangdong province, including Yunan County, Yunan District, Yuncheng District, Luoding city, and Xinxing County (Figure 1). Yunfu is located in a hilly area and is rich in mineral resources. The study area is typical of the south subtropical monsoon climate. The annual average temperature is 22.5 °C, and the annual average total precipitation is 1578.6 mm [42]. The topography of Yunfu is relatively undulating, and the overall trend is high in the northeast and low in the southwest; the forest coverage rate is 67.39%. The area is well developed in industry and mining; there is an abundance of marble and ore resources.

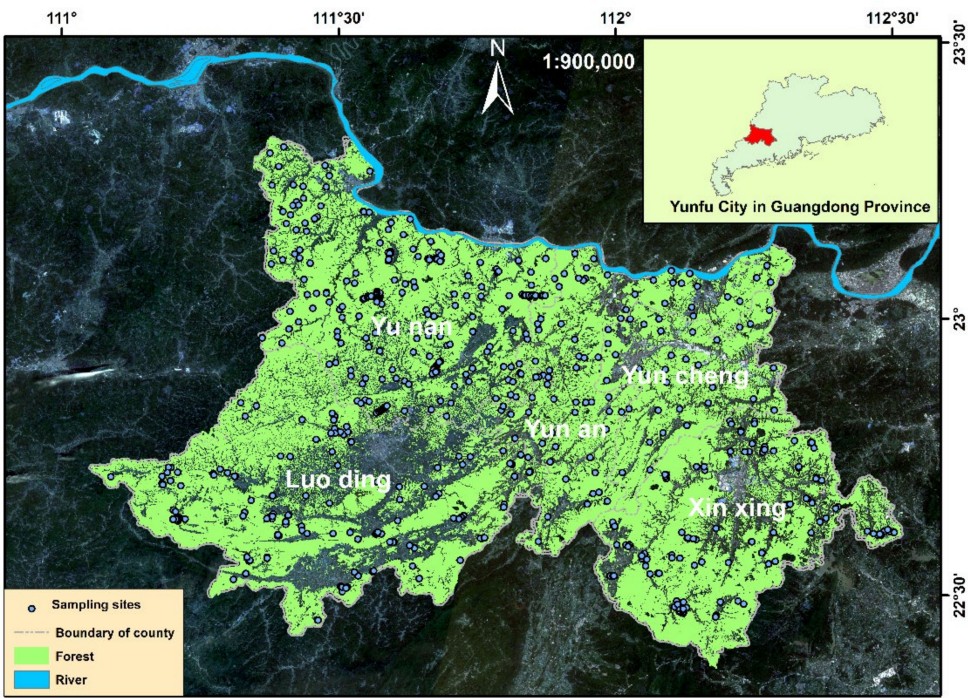

**Figure 1.** Location of the study area and sampling sites.

### 2.2. Soil Sampling and Detection

Soil samples were collected in the forest to ensure that the coverage of sampling points was large enough. A total of 727 samples (239 in Yunan County, 64 in Yunan District, 38 in Yuncheng District, 260 in Luoding city, and 125 in Xinxing County) of surface soil (0–20 cm) were collected (Figure 1). The layout of soil samples was generated according to the digital elevation model (DEM), hydrology, vegetation, land use type, and other factors, to ensure that the sample arrangement covered all types of the forest (Figure 2). These factors were obtained through satellite remote sensing interpretation, unmanned aerial vehicle field surveys, and data access. Sampling sites should be selected in areas with stable soil development conditions and less human interference. Soil samples were collected by cutting rings and were put into plastic bags and numbered. At the same time, the information on the sampling points (latitude, longitude, slope aspect, and altitude) was recorded.

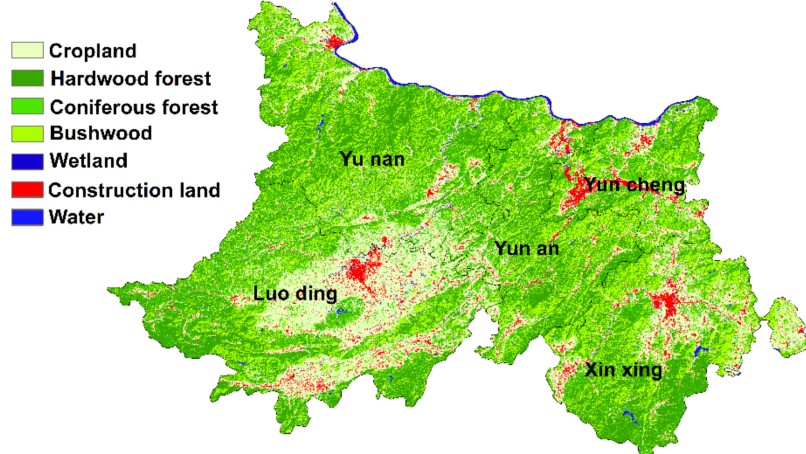

**Figure 2.** Land use types in the study area.

The soil samples were airdried and treated with a 100-mesh sieve to prepare for analysis. The contents of Cd, Pb, Cu, Zn, and Ni in the soil were determined by flame photometry after HF-HNO$_3$-HClO$_4$ digestion [43]. To better understand the soil's heavy metal content, statistical software was used to calculate the mean value and standard deviation.

### 2.3. Forest Soil Heavy Metal Pollution Assessment

In this paper, the pollution index (PI) method was applied to evaluate the level of heavy metal pollution in a certain element [44], which has been widely used in soil and water pollution assessments [45–47]. The index is mainly used to compare the measured index and evaluation standard of the heavy metal content of the soil. The formula is as follows:

$$I = \frac{Ci}{Si}$$

In the formula, *Ci* represents the content of the soil's heavy metals, *Si* represents the background value, and *i* represents a particular heavy metal element (Cd, Pb, Cu, Zn, or Ni) [48,49]. PI was classified into four grades (Table 1). The background values of heavy metals in China [50] and Guangdong province are shown in Table 2.

**Table 1.** The grades of the pollution index and potential ecological risk index.

| PI | Pollution Levels | RI | Risk Levels |
|---|---|---|---|
| PI $\leq$ 1 | Safety | RI < 150 | Low |
| 1 < PI $\leq$ 2 | Slightly contaminated | 150 $\leq$ RI < 300 | Moderate |
| 2 < PI $\leq$ 3 | Moderately contaminated | 300 $\leq$ RI < 600 | High |
| PI > 3 | Strongly contaminated | RI $\geq$ 600 | Extremely high |

**Table 2.** The statistical analysis of the heavy metal content.

| Statistical Indicators | Cd | Pb | Cu | Zn | Ni |
|---|---|---|---|---|---|
| Minimum (mg·kg$^{-1}$) | 0.00 | 0.00 | 0.00 | 0.00 | 0.00 |
| Mean (mg·kg$^{-1}$) | 0.02 | 17.05 | 10.20 | 18.14 | 7.14 |
| Maximum (mg·kg$^{-1}$) | 0.26 | 300.88 | 51.18 | 71.23 | 26.99 |
| Standard deviation (mg·kg$^{-1}$) | 0.04 | 62.29 | 13.77 | 20.18 | 8.27 |
| Guangdong (mg·kg$^{-1}$) | 0.06 | 36.00 | 17.00 | 47.30 | 14.40 |
| China (mg·kg$^{-1}$) | 0.10 | 27.00 | 23.00 | 74.00 | 27.00 |

The potential ecological risk index (RI) was usually used to assess the potential impact of heavy metal pollutants on ecosystem security and stability [51]. RI is a comprehensive index that combines heavy metal content, toxicity [52], and local background values [53,54]. The potential risk degree of heavy metals was divided by a quantitative method, which can quickly and accurately reflect the comprehensive impact of various types of heavy metal pollution [55,56]. The formula is as follows:

$$RI = \sum_{i=1}^{n} E_r^i = \sum_{i=1}^{n} T_r^i \times P_r^i \tag{1}$$

where $E_r^i$ is the RI of the single heavy metal element, $T_r^i$ is the toxicity coefficient of a certain heavy metal, $P_r^i$ is the pollution index (PI) of a certain heavy metal element. The toxicity coefficient is an important parameter of heavy metal pollution, which represents the toxicity of heavy metals to humans, animals, or plants. Based on the relevant research in western Guangdong, the toxicity coefficients of various heavy metals used in this study are as follows: Cd = 30, Pb = 5, Cu = 5, Zn = 1, and Ni = 5 [57,58]. RI is classified into four grades (Table 1).

*2.4. Spatial Pattern Analysis*

The spatial distribution of heavy metals in forest soil can provide basic information about the pollution source analysis and pollution degree evaluation [59]. In this paper, the ordinary Kriging interpolation method was used to reveal the spatial patterns of heavy metals in forest soil content and the degree of pollution. The ordinary Kriging interpolation is an optimal local estimation method that provides the best unbiased linear prediction of spatial distribution. In this study, spatial patterns of heavy metals, PI, and RI were drawn by using ArcGIS 10.3.

*2.5. Heavy Metal Source Analysis*

The principal component analysis (PCA) and Pearson's correlation were used to extract relevant data from a large number of soil samples to analyze the sources of heavy metals in forest soil. PCA and Pearson's correlation can group heavy metal pollutants that have similar sources [60,61]. PCA is mainly used for exploratory data analysis, so it is also widely used in environmental science research [1]. In previous studies, many studies analyzed the main and potential sources of heavy metals in the soil through PCA [62,63]. The analyses were performed in R 3.5.2 [64].

**3. Results**

*3.1. Content and Spatial Patterns of Heavy Metals in Forest Soils*

The mean concentrations of Cd, Pb, Cu, Zn, and Ni were 0.02 mg·kg$^{-1}$, 17.05 mg·kg$^{-1}$, 10.20 mg·kg$^{-1}$, 18.14 mg·kg$^{-1}$, and 7.14 mg·kg$^{-1}$, respectively. The mean values were lower than the background values. However, the maximum values of all heavy metals exceeded the corresponding background values in Guangdong and even in China (Table 2). Moreover, 5.37% of all forest soil samples had higher Cd content than the background values, with an average of 0.14 mg·kg$^{-1}$ (0.06 mg·kg$^{-1}$ in Guangdong and 0.14 mg·kg$^{-1}$ in China). The content of Pb in forest soil samples varied greatly, and the maximum content of Pb was 300.88 mg·kg$^{-1}$, far beyond the background value. Furthermore, the content of Cu ranged from 0 to 51.18 mg·kg$^{-1}$, and the maximum value was more than three times the background value in Guangdong province. Although the maximum content of Zn and Ni in soil samples exceeded the background values of Guangdong province, they were very close to (and still below) the background values of China.

The results of the heavy metal spatial patterns demonstrated that the distribution of heavy metal content in the forest soil varied. The highest concentrations of heavy metals were mainly found in the central part of Yunfu city, but the distribution of different metal elements was not consistent. The spatial patterns of Cd and Pb were similar, with the highest concentrations found in the southwest of Xinxing County (Figure 3a,b). The spatial distribution of Cu was different from other metal elements in the study area, and the highest values appeared in the southwest of Xinxing County and the middle of Yunan District (Figure 3c). The spatial distributions of Zn and Ni were similar, the highest values were mainly distributed in the Yunan and Yuncheng districts (Figure 3d,e).

*3.2. Correlation among Different Forest Soil Heavy Metal Elements*

The result of PCA without rotation for heavy metals showed that the first two components explained 78% of the total variance (57% for PC1 and 21% for PC2), as in Figure 3. Based on loadings of heavy metals, a clear separation of the chronologies was observed on PC2. The forest soil heavy metal elements were classified into two groups, namely Group 1 (Pb, Cd, and Cu) and Group 2 (Zn and Ni).

A correlation analysis revealed significant positive correlations among the various heavy metals (Figure 4). The content of Cu had high correlations with that of Zn (r = 0.52), Ni (r = 0.62), Pb (r = 0.56), and Cd (r = 0.61) in the soil ($p < 0.05$) (Table 3). Moreover, the content of Cd was highly correlated with Pb (r = 0.53, $p < 0.05$), and the content of Zn was highly correlated with Ni (r = 0.69, $p < 0.05$) (Table 3). The results are consistent with those of the PCA.

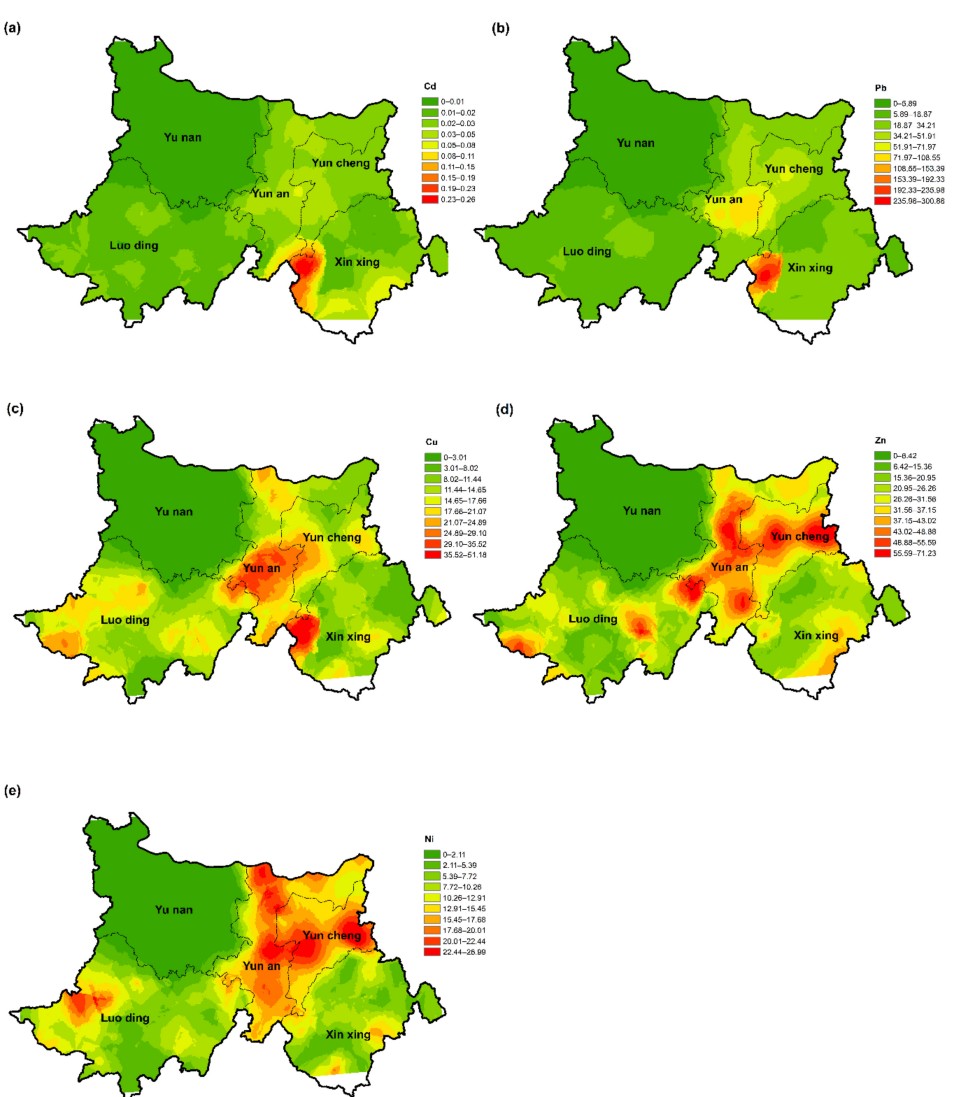

**Figure 3.** Spatial patterns of the concentrations of (**a**) Cd, (**b**) Pb, (**c**) Cu, (**d**) Zn, and (**e**) Ni in forest soil.

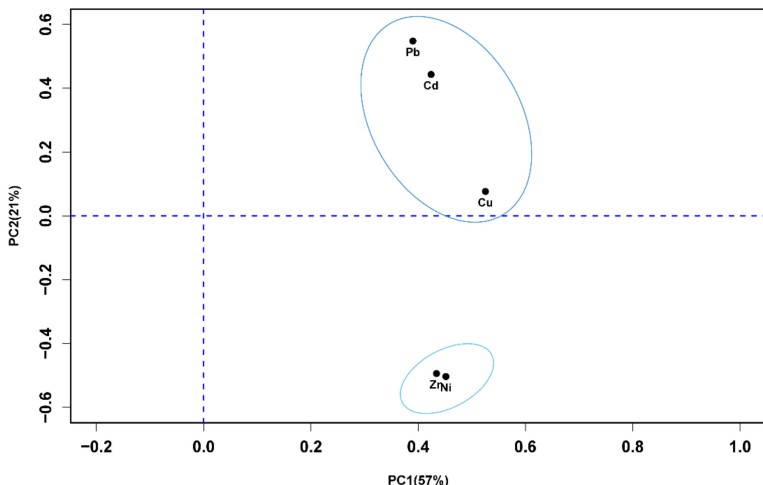

**Figure 4.** Principal component analysis of heavy metals.

**Table 3.** Correlation analysis of heavy metals.

| Elements | Cd | Pb | Cu | Zn | Ni |
|---|---|---|---|---|---|
| Cd | 1 | 0.53 ** | 0.61 ** | 0.3 | 0.31 |
| Pb | | 1 | 0.56 ** | 0.26 | 0.22 |
| Cu | | | 1 | 0.52 ** | 0.62 ** |
| Zn | | | | 1 | 0.69 ** |
| Ni | | | | | 1 |

'**' indicates the significant at level of 0.05.

### 3.3. The Spatial Pattern of the Forest Soil Pollution Index and Potential Ecological Risk Index

The spatial distributions of the pollution index indicated different contamination distribution levels of the various heavy metal elements in the study area. The Cd contamination levels were at safe levels in most of the study areas. However, the southwest of Xinxing County had areas that were moderately and strongly contaminated (PI > 1), as shown in Figure 5a. Similarly, the southwest of Xinxing County was strongly contaminated with Pb, whereas the middle of the Yunan and Yuncheng districts were only slightly and moderately contaminated (Figure 5b). The distribution of Cu pollution was relatively dispersed, except for the southwest area of Xinxing County, which was strongly contaminated. Furthermore, the central Yunan District and the western part of Luoding city had large areas of moderate and strong contamination (Figure 5c). The contamination levels of Zn were moderate and strong in most areas of the Yunan and Yuncheng districts and local areas of Luoding County (Figure 5d). The pollution levels of Ni were moderate and strong and covered most of the Yunan and Yuncheng districts and the western part of Luoding city (Figure 5e). The spatial patterns of the pollution indexes were consistent with the spatial patterns of heavy metal concentrations in Yunfu city.

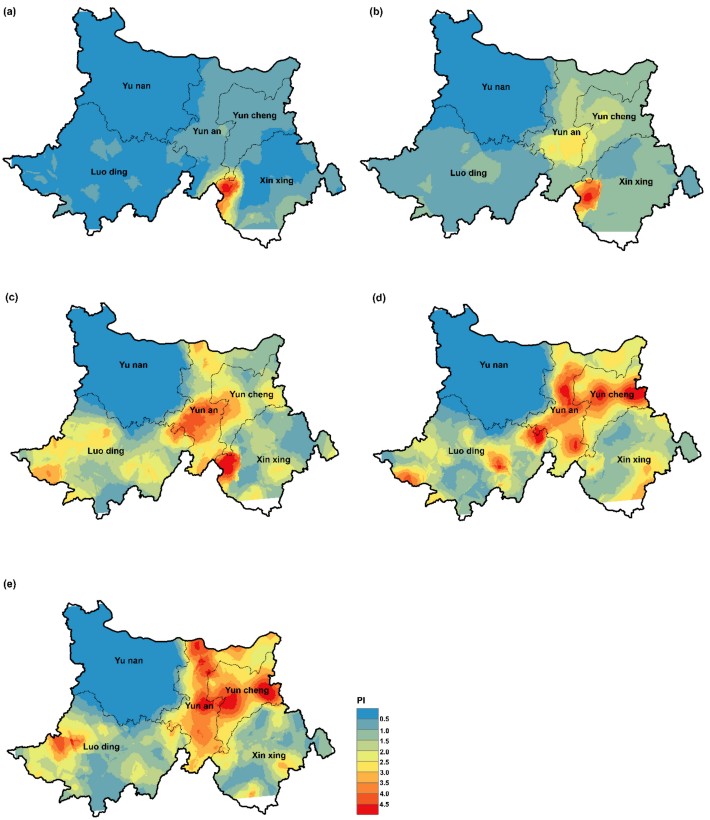

**Figure 5.** Spatial patterns of (**a**) Cd, (**b**) Pb, (**c**) Cu, (**d**) Zn, and (**e**) Ni of the forest soil pollution index.

Most of the potential ecological risks in the study area were at low levels (RI < 150). To be specific, only the southwest of Xinxing County had the highest RI and was at moderate risk. Although the RI was more than 150 in this region, its maximum value did not reach 300 (Figure 6).

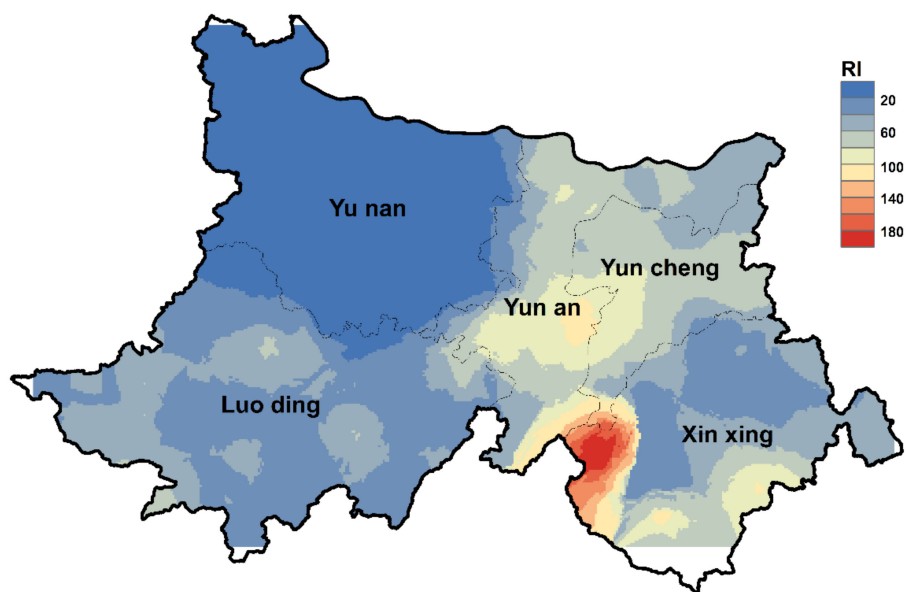

**Figure 6.** Spatial patterns of the forest soil potential ecological risk index.

## 4. Discussion

### 4.1. Analysis: Sources of Heavy Metal Pollution of Forest Soil

Based on the heavy metal content analysis of a large number of forest soil samples, the correlations of different metal elements suggested that these heavy metals may have the same source [1]. Therefore, combined with the results of PCA and the correlation analysis, we found that heavy metal elements can be divided into two groups. Given that PC1 explained 57% of the total variance with strong positive loadings for Cd, Pb, and Cu, we speculated that these three elements may have the same sources of pollution. The common sources should be waste minerals and wastewater caused by mining, which is the main industry in this area. Previous studies have demonstrated that vehicle exhaust emissions and industrial production increased Cd, Pb, and Cu in the soil [65–67]. With the acceleration of urbanization, more energy and materials were needed, and the mining industry gradually became a vital part of industrial production, which also led to more soil heavy metal pollution. It has been shown that the mining of non-ferrous metals has a significant effect on the content of Pb and Cd in the soil [68]; large amounts of tailings and waste ores transfer heavy metal elements to the forest soil through water and air [69,70]. In addition, the man-made marble slab production process could produce a large number of heavy metal pollution. Moreover, Yunfu city is the main production base of marble slabs in western Guangdong; in some areas, Pb and Cd were transferred from soil to rice and vegetables. The spatial patterns of Cd, Pb, and Cu revealed that the high concentration areas of the three elements are primarily located in the southwest of Xinxing County (Tiantang Town and Hetou Town), a major marble production area. Nevertheless, Pb and Cu in Yunan District were more likely to come from cement production [71]. As the Yunan District was one of the three major cement bases in Guangdong province, Group 1 could be ascribed to mining and cement production sources.

PC2 accounted for 21% of the total variance of Zn and Ni. The spatial distributions of these two elements were relatively consistent and were concentrated in densely populated and economically developed urban areas, such as Yunan and Yuncheng districts. Zn is a metal element widely used in the processing and manufacturing industry. For example,

aluminum alloy manufacturing [72], rubber tire manufacturing [73], and lubricating oil manufacturing [74] all need Zn as an additive element. Economically developed regions tend to have larger car ownership, which means more traffic and transportation activities [75–78]. Therefore, Zn from vehicle tires, lubricating oil, and metal shells can get into the soil and form Zn pollution. Ni is generally considered to come mainly from the soil parent material and its content varies with the soil type [76,79]; a previous study indicated that the degree of enrichment of Ni in the topsoil of Guangdong province was the highest in China [80]. However, here, the enrichment areas of Ni were mainly located in urban forests (Yunan and Yuncheng districts), and it had a high correlation with Zn. Therefore, we thought that Ni originated from the exhaust gas or waste residue discharged from the transportation industry [81]. Of course, Zn in regions other than high-value regions may have other sources. In Yunan County, we also observed that in the Zn enrichment areas, the Zn came from the heavy use of chemical fertilizers [76,82]. Yunan County is the largest agricultural and forestry production unit in the west of Guangdong province and was also one of the bases of timber forests in Guangdong province. In short, Group 2 could be interpreted as human activity factors (transportation, agricultural, and forestry production).

### 4.2. Heavy Metal Pollution of Forest Soil: Assessment

The spatial patterns of the forest soil pollution index showed that most of the forest soil in the study area was safe or had slightly contaminated levels of all heavy metals. However, the evaluation of the heavy metal pollution degree should not ignore the rich areas of heavy metal concentration [15]. Moderate to strongly contaminated degrees caused by Cd, Pb, and Cu were observed in the southwest of Xinxing County. The degrees of contamination of Pb and Cd were higher than Cu (moderate). The same results were also found in other areas where the mining industry was relatively developed [83–85]. Pb and Cd were taken out due to the development of raw ore material seeping into the soil through floating dust or tailings (with further processing of ore) [86,87]. Therefore, attention should be given to waste disposal in mineral exploitation and stone processing. Long-term heavy pollution of Pb and Cd will cause serious harm to human health [88]. Moreover, Pb showed slight to moderate contamination in the core areas of the Yunan and Yuncheng districts. Cement manufacturing is the principal resource of Pb, so dust from production should be controlled to avoid Pb pollution in a larger area [89,90]. In addition to Xinxing County, Cu had moderate to strong contamination in Yunan and Yuncheng districts and a moderate degree of contamination in Luoding County. Although forests play important roles in slowing down Cu migration across soil profiles and reducing heavy metal hazards [91], reasonable measures against industrial pollution are still needed in local areas. Zn and Ni are enriched in developed regions due to transportation and forestry management. With economic development, the degree of Zn and Ni pollution caused by transportation may increase further in local areas [31,92]. However, we can slow down this trend by reducing the use of fertilizers and pesticides in forestry operations.

Although the degrees of potential ecological risks were at medium and low levels in the study area, the higher ecological risks in local areas needed to be taken seriously. Spatial patterns across the whole study area indicated that there were moderate potential ecological risks in the southwest of Xinxing County. Similar results have been found in studies (of potential ecological risks) in other areas [76,93]. This indicates that the potential ecological risks of heavy metals from mining in the region are becoming prominent. The overlap areas between the spatial patterns of PI and RI were mainly distributed in the southwest of Xinxing County. Therefore, we realized that the higher ecological risk level was mainly caused by Cd and Pb. The potential ecological hazards of Pb and Cd were extremely high. Some researchers have found that the potential ecological risk index caused by Pb and Cd in the surrounding soil was at a medium (or higher) level in other areas with similar industries [94,95]. Other elements showed a light grade of ecological risk. This might be because Zn and Ni could be combined with other elements in the soil [96,97], and returned to the geological cycle of elements, with less potential impact on the ecological

environment [98–100]. However, with further societal developments, Zn and Ni may become potential polluters. In the southwest of Xinxing County, more attention should be given to Cd and Pb to control heavy metal pollution of soil in the future.

## 5. Conclusions

In this study, we analyzed the spatial distributions of heavy metal elements and the spatial patterns of pollution characteristics of forest soil and analyzed the main sources of heavy metal elements in western Guangdong. The mean concentrations of these five metals were less than their corresponding critical values, but they were concentrated locally. The spatial patterns of heavy metal concentrations in forest soil were not uniform, and there were obvious differences among different regions in the study area. Cd, Pb, and Cu were mainly enriched in the southwest of Xinxing County, while Zn and Ni were enriched in most areas of the Yunan and Yuncheng districts. The results of PCA and the correlation analysis indicated that heavy metals in soil could be divided into two groups with diversified sources. Group 1 represents sources of mining and cement production (Cd, Pb, and Cu), while Group 2 involved mixed sources of transportation and chemical fertilizers (Zn and Ni). We calculated the pollution index and potential ecological risk index to reveal the pollution characteristics. The results indicated that the pollution levels of heavy metals in the enrichment area were moderate to strong, with safe or slightly contaminated levels in most study areas. The potential ecological risk was at a lower level in the study area, but it is necessary to manage heavy metals in the forest soil in southwest Xinxing County. In conclusion, heavy metal pollution in the forest west of Guangdong was affected by human activities, which have led to a significant increase in potential ecological risks.

**Author Contributions:** Conceptualization and methodology, J.K., X.D.; formal analysis, J.K., X.D. and J.H.; investigation, H.M.; data curation, X.L., Z.D.; writing—original draft preparation, J.K., X.D.; writing—review and editing, X.L., J.H.; visualization, J.K.; supervision, X.D. All authors have read and agreed to the published version of the manuscript.

**Funding:** This research was funded by the Guangdong Forestry Science and Technology Plan of China (grant no. 2019-07) and by the Guangdong Natural Science Foundation (grant no. 2019B121202007).

**Data Availability Statement:** Not applicable.

**Acknowledgments:** We would like to thank Zhongrui Zhang, Yao Jiang from Guangdong Academy of Forestry sciences for their kindly help in field work. We also thank forestry department of Yunfu Municipal Government (Yunan County, Yunan District, Yuncheng District, Luoding city, and Xinxing County) for their support.

**Conflicts of Interest:** The authors declare no conflict of interest.

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
