# Peer review of "Characteristics and Risk of Forest Soil Heavy Metal Pollution in Western Guangdong Province, China"

_forests, doi:10.3390/f13060884_

Round 1
Reviewer 1 Report
Manuscript Number: forests-1691232 (MDPI)
Title: Characteristics and risk of forest soil heavy metal pollution in Western Guangdong Province, China
The paper contain a lot of data concerning heavy metal pollution in Western Guangdong Province in China. I found this article interesting and significant for the field of study. I think manuscript in the present form fits the scope of a journal. Although, the article is prepared fairy well, I noticed some inaccuracies and shortcomings. I can recommend the major revision.
Comments:
- The biggest disadvantage of the article is too general Introduction that do not focus on the problem and contain very general knowledge on heavy metals and soil. I think authors should omit this information and provide more details concerning on the threats of heavy metal for the studied soils, the potential source of pollution and the data from the previous studies (art of study). However, the paragraph (lines 50-65( is very good!
- Another problem. The included figures have low quality… I do not mean only the size of the figure but also the content. I think the current form of most of figure is too simple.
Figures 2, 5 and 6: I suggest to add the names of research areas, localities, etc. on the map.
- Authors based on the formulas for Pollution Index and Potential Ecological Risk. I suggest to cite original source of those indices and/or add some reviews that contain and discuss the formulas.
- Results:
Line 141-142, this sentence is not relevant.
Lines 154-163: this paragraph is not consistent in content. Sometimes the authors write that the distribution of heavy metals is varied (e.g. 155) and once that it is uniform (e.g. 160)... what is the truth?
- Discussion:
4.1. What is the source of heavy metal exactly (line 212)? What the heavy metal pollution came from exactly? Mines of what? What name? Please provide previous research (line 218).
4.2. Line 271-273: Please provide more source, more international literature.
- Conclusion:
Line 280: Please refer to the results: whether the distribution is uniform or varied? In general, the conclusions should correspond to the assumed goals, and to the assumed issue. In the present form the conclusions have a character of abstract….
- References:
The serious problem with the literature and references is as follow: too much papers of local authors! Please provide more international references! After all, there are many reviews and study cases of known international authors. A lot of international authors outside of China deal with pollution indices. The references should show the knowledge of international recent literature.
Reviewer 2 Report
Remarks
The introduction needs of revision in order to be more focused. The explanations not supported by references should be avoided. There is no clear evidence, why the study region is selected for the study – the importance of the region and need of special attention are underlined, but this is not well supported with information about surrounding sources of pollution and for the readers this is not clear.
The quality of fig.1 needs of improvement
The methodology is well described. More details are needed for the sampling design – while a large number of samples from superficial soil (0-20 cm) are collected there is no clear differentiation about the points and criteria applied. The distance from the sources of pollution is an important factor, which is not considered, but could provide essential information about the behaviour of pollutants and their spatial distribution.
There is a lack of information about the soil types, their textural composition and pH, which play important role for the heavy metals’ behaviour and overall conditions of forest soils in case of pollution.
The quality of fig. 2, fig.3 and fig. 4 …needs of improvement
The discussion of the results is not well linked to the information in maps provided – the name of cities and localities with higher risks are mentioned and discussed, while they are not linked to the information in maps. This makes the study rather limited and refers it to a study with regional interest in terms of information and problems discussed.
Conclusions are not well supported by the results, while the sources of pollutions are not even mentioned in the text but the general conclusion is based on it.
Round 2
Reviewer 1 Report
The article has been corrected according to the suggestions. All comments have been considered. I think that the article is ready for publication.
Reviewer 2 Report
Dear authors,
most of the remarks and comments in the review are considered.
The MS could be accepted for publishing.